

# Geochemical characteristics of dissolved heavy metals in Zhujiang River, Southwest China: spatial-temporal distribution, source, export flux estimation, and a water quality assessment

Jie Zeng[1], Guilin Han[1], Qixin Wu[2] and Yang Tang[3]

[1] School of Scientific Research, China University of Geosciences (Beijing), Beijing, China
[2] Key Laboratory of Karst Environment and Geohazard, Ministry of Land and Resources, Guizhou University, Guiyang, China
[3] State Key Laboratory of Environmental Geochemistry, Institute of Geochemistry, Chinese Academy of Sciences, Guiyang, China

## ABSTRACT

To investigate the sources and spatial-temporal distribution of dissolved heavy metals in river water, and to evaluate the water quality, a total of 162 water samples were collected from 81 key sampling points in high and low flow seasons separately in the Zhujiang River, Southwest China. Ten dissolved heavy metals (V, Cr, Mn, Co, Ni, Cu, Mo, Cd, Ba, and Pb) in the Zhujiang River water exhibit little variation at temporal scale, but vary with a significant spatial heterogeneity. Furthermore, different metals present different variation trends along the main channel of the Zhujiang River. Our results suggest that Ba (14.72 $\mu g \ L^{-1}$ in low flow season and 12.50 $\mu g \ L^{-1}$ in high flow season) and Cr (6.85 $\mu g \ L^{-1}$ in low flow season and 7.52 $\mu g \ L^{-1}$ in high flow season) are consistently the most abundant metals in the two sampling periods. According to the water quality index (WQI values ranged from 1.3 to 43.9) and health risk assessment, metals investigated in Zhujiang River are below the hazard level (all hazard index (HI) < 1). Application of statistical approaches, including correlation matrix and principal component analysis (PCA), identify three principal components that account for 61.74% of the total variance, the results conclude that the anthropogenic heavy metals (V, Cr, Ni, and Cu) are greatly impacted by the dilution effect, and the heavy metals in Zhujiang River are mainly presented a natural sources signature from the perspective of entire basin. Moreover, our results reveal that the estimated export budget of several heavy metals including V (735.6 t year$^{-1}$), Cr (1,561.1 t year$^{-1}$), Ni (498.2 t year$^{-1}$), and Mo (118.9 t year$^{-1}$) to the ocean are higher than the world average.

Corresponding author
Guilin Han, hanguilin@cugb.edu.cn

## INTRODUCTION

Along with the rapid development of social economy, the contamination levels and environmental health effects of heavy metals aroused great public concern in the world due to their toxicity, persistence, bioaccumulative nature, and mostly negative impact on living organisms (*Cameron, Mata & Riquelme, 2018*; *Farahat & Linderholm, 2015*; *Wilbers et al., 2014*; *Zaric et al., 2018*). There are two main sources of heavy metals in the environment that significantly impact the biogeochemical cycling of heavy metals (*Li & Zhang, 2010*). One is natural sources, such as bedrock weathering and volcanism, which are controlled by geology and lithology (*Krishna, Satyanarayanan & Govil, 2009*; *Li & Zhang, 2010*); the other is anthropogenic activities, including mining, metal smelting and refining, energy producing and consuming, and waste incineration (*Liu et al., 2013*; *Meng et al., 2016*). We can identify the heavy metal sources (natural and/or anthropogenic) by detecting their contents and distribution in the ecosystem.

Studies about environmental heavy metals have a long history (*Zhang et al., 2009*). These studies focused on various scales and different ecosystems, especially the fluvial system (*Iwashita & Shimamura, 2003*; *Li & Zhang, 2010*; *Wang et al., 2017*). Hitherto, numerous studies regarding heavy metal (or trace element) compositions and their effects on fluvial environment have been published in different countries (*Gaillardet, Viers & Dupré, 2014*; *Iwashita & Shimamura, 2003*; *Thévenot et al., 2007*; *Tripathee et al., 2016*), including China (*Li, Li & Zhang, 2011*; *Wang et al., 2017*; *Xiao, Jin & Wang, 2014*; *Zhang & Zhou, 1992*). As the second largest river in China and the major river discharging into the South China Sea (SCS) (*Xu & Han, 2009*), several researches have been performed to examine heavy metal compositions of water and sediment in the Zhujiang River basin (*Liu et al., 2017*; *Niu et al., 2009*; *Ouyang et al., 2004*; *Zhang et al., 2018*; *Zhen et al., 2016*). However, the spatial-temporal distribution, source, health risk, and export flux of dissolved heavy metals in Zhujiang River from a whole basin perspective, have not been investigated systematically up to now.

This study is conducted as a survey on water geochemistry characteristics of dissolved heavy metals in the Zhujiang River from a whole basin perspective. The aims are to: (1) analyze the spatial-temporal distribution of 10 dissolved heavy metals from the upper reaches to the lower reaches, (2) explore the possible anthropogenic and/or natural sources of these heavy metals, (3) define the water quality and health risk of dissolved heavy metals, and (4) estimate the export flux of dissolved heavy metals to SCS roughly. The results would greatly help manage and protect the water resources, provide water resources guarantee for socio-economic development within the whole basin, and deliver some support for the assessment of heavy metal fluxes input to the ocean.

## MATERIALS AND METHODS

### Study area

The Zhujiang River (Pearl River) basin is located in Southwest China (21°31′–26°49′N, 102°14′–115°53′E), which is the largest river flowing into the SCS with a cover area of

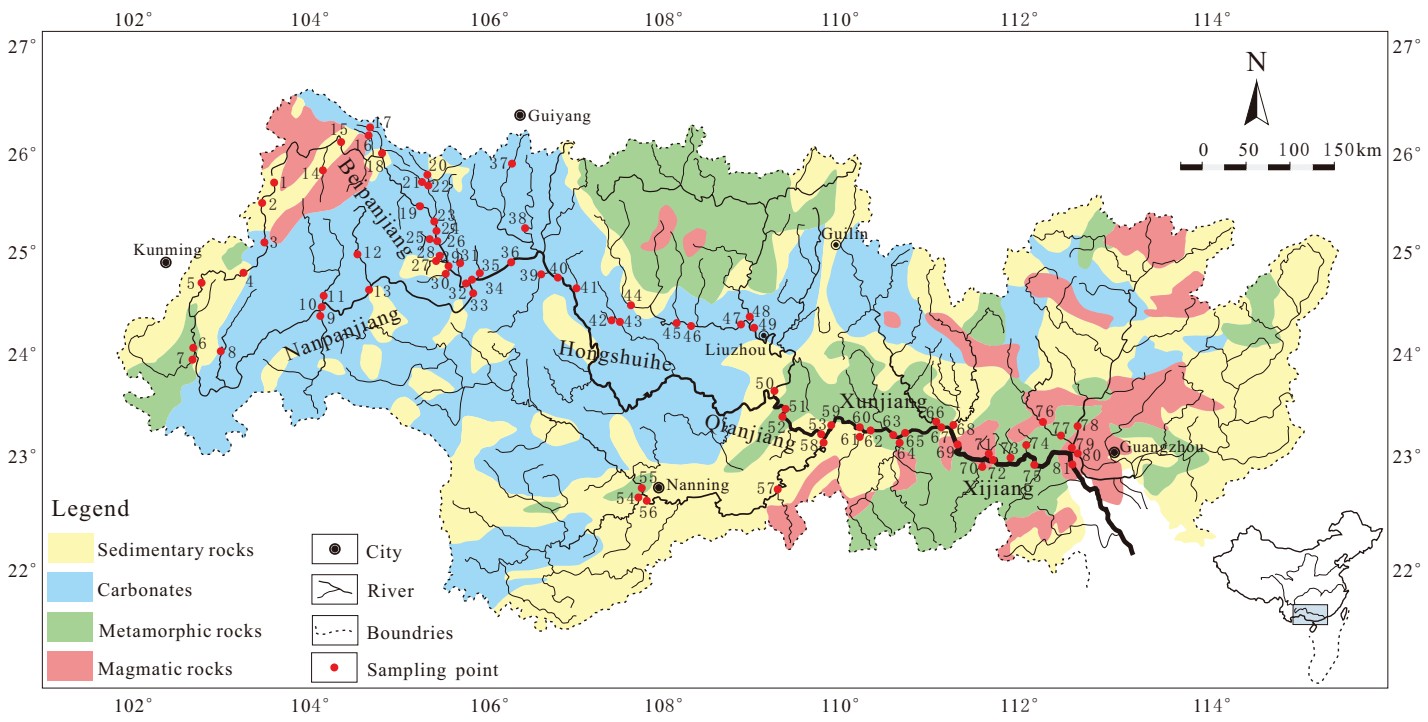

**Figure 1 Sketch map showing the lithology and sampling locations of the Zhujiang River system.** The base map was modified from *Han et al. (2018)*.

$4.5 \times 10^5$ km$^2$ and provides major water source for more than 30 million local population of southern China (*Li et al., 2019*; *Zhen et al., 2016*). The Zhujiang River flows through Yunnan, Guizhou, Guangxi, and Guangdong province with the elevation decreases from northwest to southeast (Fig. 1). Five principal reaches are defined along the main channel of Zhujiang River: the Nanpanjiang River (NPR), the Hongshuihe River (HSR), the Qianjiang River (QJR), Xunjiang River (XUR), and Xijiang River (XJR) (Fig. 1). Accordingly, the tributaries can also be divided into five parts. In addition, the significant tributary (Beipanjiang River, BPR) located in the upstream area, is discussed as an individual tributary here. The Zhujiang River basin is characterized by tropical to subtropical monsoon climate, with a mean annual temperature varying from 14 to 22 °C. The average annual precipitation ranges from 1,200 to 2,200 mm, where a majority of annual rainfall (~80%) occurs during the wet season (April–September) (*Han et al., 2018*; *Li et al., 2019*). The Zhujiang River basin consists of various source rocks (Fig. 1). Permian and Triassic carbonate rocks with intercalated coal-bearing formations are extensively distributed in the mid-upper river reaches with an area account for 44% of the whole basin area. Jurassic detrital sedimentary rocks are distributed sporadically in the river source area and are intercalated fragmentarily in the middle and lower river reaches. The mid-lower reaches consist mostly of Precambrian metamorphic rocks and magmatic rocks (*Han et al., 2018*; *Li et al., 2019*, *2000*). There are 24 large reservoirs/dams (with capacity exceed $10^8$ m$^3$) and 212 medium reservoirs/dams (capacity from $10^7$ to$10^8$ m$^3$) located in the mid-lower reaches of the Zhujiang River (*Han et al., 2018*).

## Sample collection and analysis

Two sampling campaigns from 81 sites represent varying hydrologic period settings throughout the Zhujiang River during July 2014 (high flow season) and January 2015 (low flow season) are conducted (Fig. 1). And 9, 3, 3, 5, and 5 sites are located in NPR, HSR, QJR, XUR, and XJR, while 5, 7, 7, 7, 11, and 19 sites are located in the corresponding tributaries of NPR, HSR, QJR, XUR, XJR, and BPR. More detailed information of the sampling points could be found in Table S1. Thus, a total of 162 river water samples are collected at a depth of approximate 15 cm. After collection, the water samples are immediately filtered in the field through 0.22 μm Millipore membrane filters. Water samples for heavy metals analysis are acidified to pH < 2 with ultra-purified $HNO_3$ and then sealed in a pre-cleaned polyethylene bottles and stored in a dark refrigerator before analysis. The pH, electrical conductivity (EC), dissolved oxygen (DO), and temperature (T) are immediately measured by a multi-parameter meter (MultiLine 3320; WTW, Weilheim, Upper Bavaria, Germany) in the field. Ten heavy metals, including V, Cr, Mn, Co, Ni, Cu, Mo, Cd, Ba, and Pb are analyzed by inductively coupled plasma-mass spectrometry (Elan DRC-e; Perkin Elmer, Waltham, MA, USA) at the Institute of Geographic Sciences and Natural Resources Research, Chinese Academy of Sciences. Standard reference materials (GSB 04-1767-2004) are used to perform the method validation and quality control. All the samples and standards are analyzed in batches with a procedural blank. Recovery percentage ranges between 90.0% and 110.4%. Relative standard deviations for heavy metals are ~±5%.

## Statistical analysis and assessment methods

### Multivariate analysis

Statistical approaches, including correlation matrix and principal component analysis (PCA) are used to analyze the dataset to acquire descriptive statistics and explore the possible sources of the selected dissolved heavy metals. PCA is the most common multivariate statistical method to explore associations and origins of heavy metals (*Loska & Wiechuła, 2003*). PCA is generally via reducing the dimensionality of the dataset to several influencing factors while trying to preserve the relationships presented in the original data, a fact which commonly occurs in hydrochemistry (*Li, Li & Zhang, 2011*; *Wang et al., 2017*). The suitability of the dataset for factor analysis is checked by Kaiser-Meyer-Olkin (KMO) and Bartlett's sphericity test ($p < 0.001$) (*Li, Li & Zhang, 2011*). Before performing the PCA, each variable is first normalized to avoid numerical ranges of the original variables by $z$-scale transformation (*Chen et al., 2007*; *Wang et al., 2017*). Then, PCA with varimax rotation is run. Only components with eigenvalues exceeding one after rotation are extracted. All of the data processes are performed using the Microsoft Office 2010 and the statistical software package SPSS 21.0 for Windows.

### Water quality index

The water quality index (WQI), a powerful tool that reflects the integrated influence of different water quality variables (heavy metals in this study), present a comprehensive picture of river water quality (*Wang et al., 2017*; *Xiao, Jin & Wang, 2014*). V is excluded

in the WQI calculations due to the lack of official drinking water standard. The WQI is calculated as follows:

$$\text{WQI} = \sum \left( w_i \times \left( \frac{C_i}{S_i} \right) \right) \times 100 \tag{1}$$

where $w_i$ is the weight of each heavy metal $i$ and represents the relative importance of different water quality variables in the overall quality of water for drinking, and is depend on the foundation of the eigenvalues for each principal component and factor loading for each parameter from the PCA results. $C_i$ is the heavy metal concentration of water samples ($\mu g\ L^{-1}$), and $S_i$ is the Chinese Drinking Water Guideline (GB 5749-2006) for each heavy metal ($\mu g\ L^{-1}$). The obtained WQI values can be defined as five classifications: excellent water quality ($0 \leq \text{WQI} < 50$), good water quality ($50 \leq \text{WQI} < 100$), poor water quality ($100 \leq \text{WQI} < 200$), very poor water quality ($200 \leq \text{WQI} < 300$), and WQI > 300 represents water that is unsuitable for drinking (*Meng et al., 2016*; *Wang et al., 2017*).

### Health risk assessment

To quantitatively evaluate health risks, the Hazard quotient (HQ) and hazard index (HI), which are suggested by the US EPA and have been commonly used for the river water risk assessments in previous studies (*Meng et al., 2016*; *Wang et al., 2017*), are calculated in this study. Ingestion and dermal absorption are two main pathways potentially expose to heavy metals in water for humans beings (*De Miguel et al., 2007*). HQ is the ratio between exposure via each pathways and the reference dose (RfD). HI is the sum of the HQs for each heavy metal from both the ingestion and dermal pathways to assess the total potential non-carcinogenic risk of individual metals. If the HQ or HI exceeds 1, non-carcinogenic risk/adverse effects on human health are a concern and the further study is necessary. In contrast, there are no deleterious effects when HQ/HI is smaller than 1 (*Wang et al., 2017*). The HQ and HI are calculated as follows:

$$\text{ADD}_{\text{ingestion}} = \frac{(C_w \times \text{IR} \times \text{EF} \times \text{ED})}{(\text{BW} \times \text{AT})} \tag{2}$$

$$\text{ADD}_{\text{dermal}} = \frac{(C_w \times \text{SA} \times K_p \times \text{ET} \times \text{EF} \times \text{ED} \times 10^{-3})}{(\text{BW} \times \text{AT})} \tag{3}$$

$$\text{HQ} = \frac{\text{ADD}}{\text{RfD}} \tag{4}$$

$$\text{RfD}_{\text{dermal}} = \text{RfD} \times \text{ABS}_{\text{GI}} \tag{5}$$

$$\text{HI} = \sum \text{HQs} \tag{6}$$

where $\text{ADD}_{\text{ingestion}}$ and $\text{ADD}_{\text{dermal}}$ are the average daily doses by ingestion and dermal absorption ($\mu g\ kg^{-1}\ day^{-1}$), respectively; $C_w$ is the heavy metal concentration of water samples ($\mu g\ L^{-1}$); BW is the average body weight for adults and children (kg); IR is the ingestion rate (L $day^{-1}$); EF is the exposure frequency (days $year^{-1}$); ED is the exposure duration (years); AT is the average time (days); SA is area the exposed skin ($cm^2$);

Table 1 Concentrations of dissolved heavy metals (µg L$^{-1}$), pH, electric conductivity (µS cm$^{-1}$), dissolved oxygen (mg L$^{-1}$), and temperature (°C) in the Zhujiang River, China.

| | Low flow season | | | | | | High flow season | | | | | | Drinking water guidelines | | | Source area of Yangtze River[e] |
|---|---|---|---|---|---|---|---|---|---|---|---|---|---|---|---|---|
| | Min | Max | Mean | SD | Median | K–S test[a] | Min | Max | Mean | SD | Median | K–S test[a] | China[b] | WHO[c] | US EPA[d] | |
| V | 1.51 | 3.00 | 2.11 | 0.32 | 2.13 | 0.758 | 1.96 | 3.53 | 2.66 | 0.36 | 2.59 | 0.469 | | | | 0.23 |
| Cr | 2.51 | 12.00 | 6.78 | 1.79 | 6.85 | 0.864 | 1.82 | 14.96 | 7.45 | 2.84 | 7.52 | 0.709 | 50 | 50 | 100 | 0.26 |
| Mn | 0.15 | 267.33 | 8.47 | 39.99 | 0.45 | 0.000 | 0.11 | 134.49 | 5.84 | 17.95 | 0.40 | 0.000 | 100 | 400 | | 2.53 |
| Co | 0.03 | 1.57 | 0.15 | 0.18 | 0.11 | 0.000 | 0.04 | 0.51 | 0.12 | 0.08 | 0.11 | 0.001 | 1,000 | | | 0.24 |
| Ni | 0.47 | 37.35 | 4.10 | 6.33 | 2.39 | 0.000 | 0.47 | 49.03 | 4.43 | 7.50 | 2.08 | 0.000 | 20 | 70 | | 0.18 |
| Cu | 0.33 | 115.73 | 3.93 | 13.83 | 0.90 | 0.000 | 0.24 | 136.80 | 8.24 | 23.34 | 0.77 | 0.000 | 1,000 | 2,000 | 1,300 | 0.63 |
| Mo | 0.11 | 95.75 | 1.89 | 10.58 | 0.63 | 0.000 | 0.13 | 0.96 | 0.49 | 0.21 | 0.44 | 0.134 | 70 | | | 0.72 |
| Cd | 0.02 | 2.09 | 0.09 | 0.23 | 0.04 | 0.000 | 0.02 | 1.36 | 0.06 | 0.15 | 0.03 | 0.000 | 5 | 3 | 5 | 0.02 |
| Ba | 7.99 | 46.79 | 16.59 | 7.12 | 14.72 | 0.000 | 4.00 | 48.40 | 14.22 | 7.59 | 12.50 | 0.001 | 700 | 700 | 2,000 | |
| Pb | 0.03 | 0.60 | 0.07 | 0.07 | 0.06 | 0.000 | 0.02 | 0.45 | 0.06 | 0.06 | 0.04 | 0.000 | 10 | 10 | 15 | 0.76 |
| pH | 7.0 | 8.8 | 7.9 | 0.4 | 7.9 | 0.546 | 6.4 | 8.4 | 7.7 | 0.4 | 7.7 | 0.659 | 6.5–8.5 | | | |
| EC | 76.0 | 602.0 | 354.6 | 123.3 | 353.0 | 0.606 | 90.0 | 533.0 | 307.7 | 114.2 | 319.0 | 0.627 | | | | |
| DO | 6.1 | 11.8 | 8.7 | 1.0 | 8.8 | 0.976 | 4.9 | 12.4 | 7.5 | 1.3 | 7.3 | 0.316 | | | | |
| T | 10.2 | 26.6 | 16.4 | 2.7 | 16.8 | 0.096 | 18.0 | 36.0 | 26.5 | 4.2 | 27.3 | 0.425 | | | | |

Notes:
[a] Kolmogorov–Smirnov test.
[b] Chinese drinking water standards (GB 5749-2006).
[c] WHO (2006) drinking water guidelines.
[d] US EPA (2003) drinking water standards.
[e] Zhang & Zhou (1992).

ET is exposure time (h day$^{-1}$); $K_p$ is the dermal permeability coefficient in water of individual metals (cm h$^{-1}$); RfD is the corresponding reference dose (µg kg$^{-1}$ day$^{-1}$); and ABS$_{GI}$ is the gastrointestinal absorption factor (dimensionless) (Wang et al., 2017; Wu et al., 2009). The corresponding parameters are obtained from the United States Environmental Protection Agency (EPA) (2004).

## RESULTS

### Kolmogorov–Smirnov test of data

The Kolmogorov–Smirnov test, as a non-parametric test, is generally used to data analysis when the sample size is small. Thus, the normal distribution test of our data is using Kolmogorov–Smirnov (K–S) statistics (Table 1). The test results show that the V, Cr, pH, EC, and DO are normally distributed during low flow season, and the V, Cr, Mo, pH, EC, DO, and temperature approach a normal distribution during high flow season. The K–S test results and large standard deviation values for the remaining dissolved heavy metals indicatives of that the average concentration might have been seriously impacted by the outliers, which related to water samples with tremendously high or low values. Thus, the median concentrations instead of arithmetic means for these heavy metals are used in our calculations. In addition, based on the K–S test results, the median concentrations of all studied dissolved heavy metals are used for comparisons, although the limited values in the guidelines are widely given as arithmetic means.

## PHYSICOCHEMICAL PARAMETERS

The statistics values of water quality parameters (pH, EC, DO, and T) in the water samples from the Zhujiang River are given in Table 1. The average pH values are 7.9 (7.0–8.8) in low flow season and 7.7 (6.4–8.4) in high flow season, presenting slightly alkaline characteristics. The EC values of water samples vary from 76.0 to 602.0 µS cm$^{-1}$, with an average of 354.6 µS cm$^{-1}$ in low flow season, and range from 90.0 to 533.0 µS cm$^{-1}$, with an average of 307.7 µS cm$^{-1}$ in high flow season, respectively. The average DO values are 8.7 mg L$^{-1}$ in low flow season and 7.5 mg L$^{-1}$ in high flow season, respectively. Overall, the pH, EC, and DO values are consistently higher in low flow season than in high flow season. However, since the river water is continuously heated by sunlight, the water temperatures in high flow season (26.5 °C, with a range of 18.0–36.0 °C) are higher than in low flow season (16.4 °C, with a range of 10.2–26.6 °C). The variations of water temperature during the same season are primary caused by the differences of sampling time of day, that is, the water temperatures in morning are generally lower than noon.

## HEAVY METALS CONTENT

The descriptive statistics of studied dissolved heavy metals in Zhujiang River are presented in Table 1. The concentration (median) of 10 heavy metals in low flow season is decreased in the following sequence, Ba (14.72 µg L$^{-1}$), Cr (6.85 µg L$^{-1}$), Ni (2.39 µg L$^{-1}$), V (2.13 µg L$^{-1}$), Cu (0.90 µg L$^{-1}$), Mo (0.63 µg L$^{-1}$), Mn (0.45 µg L$^{-1}$), Co (0.11 µg L$^{-1}$), Pb (0.06 µg L$^{-1}$), Cd (0.04 µg L$^{-1}$). Whereas the declined sequence of these heavy metals in the high flow season is as follows, Ba (12.50 µg L$^{-1}$), Cr (7.52 µg L$^{-1}$), V (2.59 µg L$^{-1}$), Ni (2.08 µg L$^{-1}$), Cu (0.77 µg L$^{-1}$), Mo (0.44 µg L$^{-1}$), Mn (0.40 µg L$^{-1}$), Co (0.11 µg L$^{-1}$), Pb (0.04 µg L$^{-1}$), Cd (0.03 µg L$^{-1}$). Ba and Cr are consistently the most abundant metals in the two sampling periods.

## SPATIAL DISTRIBUTION OF HEAVY METALS

### Distribution of heavy metals along the main stream

The distribution of each individual heavy metal along the main stream exhibits an extensive variation (Fig. 2). In the main stream water body, V concentration present a gradually increasing trend from upstream to downstream overall, with a higher concentration in the middle of the NPR reach. In contrast, Cr and Co exhibits a slowly decreasing trend from upstream to downstream, but perform a higher concentration in the middle of the NPR reach. Cu concentration is maintained at a low level, and present a very high value for only a few sampling points (10 times more than other points). Mo concentration gradually increase in the NPR reach and began to decline after entering the HSR reach, until the QJR reach become stable. There is a great fluctuation of Ba concentration in the upstream reach, with the highest concentration in the middle of NPR reach, and subsequently decrease to the lowest value in QJR reach, and then exhibit a slowly rising trend in the downstream reach (XUR and XJR). Pb concentration is fluctuated in a relatively wider range in NPR and HSR reach, and tended to stabilize in QJR to XJR reach. Mn, Ni, and Cd display obviously fluctuation along the main stream, with no significant pattern of distribution. Moreover, there are 24 large reservoirs/dams

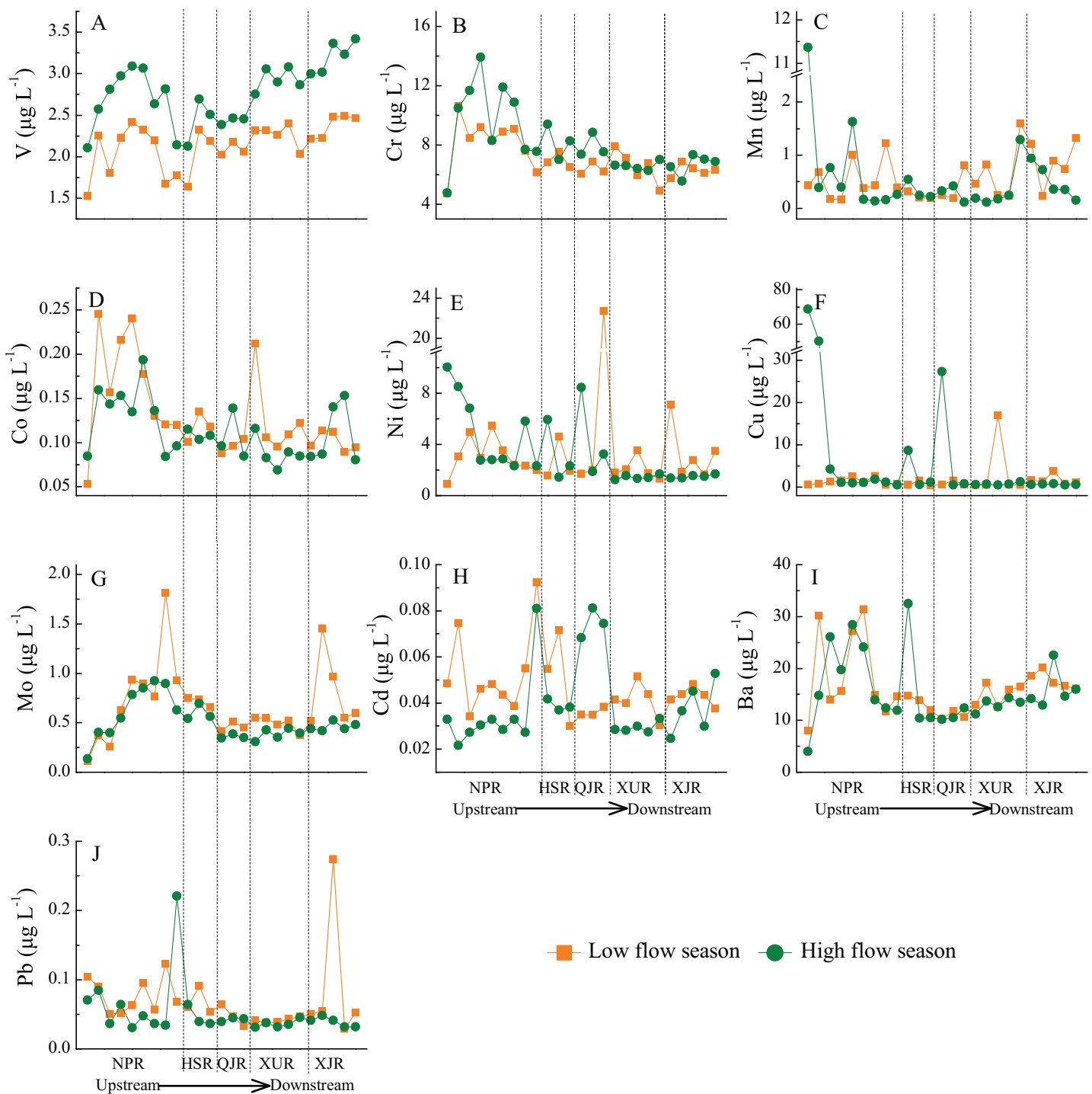

**Figure 2 Spatial-temporal variations of 10 heavy metals in 25 sampling sites along the main stream from Zhujiang River.** NPR, Nanpanjiang River; HSR, Hongshuihe River; QJR, Qianjiang River; XUJ, Xunjiang River; XJR, Xijiang River. The vertical black dash line divides the different river reaches. (A) V concentrations. (B) Cr concentrations. (C) Mn concentrations. (D) Co concentrations. (E) Ni concentrations. (F) Cu concentrations. (G) Mo concentrations. (H) Cd concentrations. (I) Ba concentrations. (J) Pb concentrations.

located in the mid-lower reaches along the main channel of the Zhujiang River (*Han et al., 2018*). The suspended particles, gravels that absorbed heavy metals would be separated from the water physically as water flow slows down in the reservoirs (*Li et al., 2008*), which could be considered as a potentially factor for the spatial variation of the dissolved heavy metals, and may result in the variation of concentration of some dissolved heavy metals in the mid-lower reaches.

### Distribution of heavy metals in tributaries

The distribution of each individual heavy metals in tributaries of different river reaches is exhibited in Fig. 3. Compare the median concentrations of heavy metals in tributaries, the V concentration is lower in the upstream tributaries than the downstream tributaries (except T-NPR), while Cr and Co are higher in the upstream tributaries than the downstream tributaries. It is consistent with the V, Cr, and Co concentration distribution in the main stream. Inconsistent with the main stream, Ni concentration present a similar distribution pattern to Cr and Co in the tributaries of each reaches, which is higher in the upstream tributaries than the downstream tributaries. In addition, Mn concentration shows that the upstream tributaries are lower than the downstream tributaries, and the fluctuated range is obviously increased in the tributaries of the downstream region. The remaining heavy metals (Cu, Mo, Ba, Pb, and Cd) exhibit an extensive variation in the tributaries of each reaches, with no significant pattern, but it should be noted that the fluctuation of several heavy metals (Mo, Pb, and Cd) in the downstream tributaries are relatively more intense than in the upstream tributaries.

## SEASONAL VARIATION OF HEAVY METALS

On the seasonal scale, only V concentration exhibit a significant seasonal variation in both of main stream and tributaries, which is consistently higher in high flow season than in low flow season (Figs. 2 and 3). Along the main stream, Cr concentration is also higher in high flow season than in low flow season (except for a few points in the midstream and downstream) (Fig. 2). In contrast, Co, Mo, Ba, Pb, and Cd concentrations are higher in low flow season than in high flow season for most of sampling sites. However, there are no significant patterns of the seasonal variation of Mn, Ni, and Cu concentrations along the main stream (Fig. 2). For tributaries, the seasonal variation patterns of the concentrations of all heavy metals (except V) are not obvious (Fig. 3). Generally, the concentration of dissolved heavy metals would be reduced by the dilution effect during the high flow season, which is controlled by river flow (*Li & Zhang, 2010*; *Olías et al., 2004*). Therefore, the Co, Mo, Ba, Pb, and Cd concentrations are higher in low flow season than in high flow season at most of sampling sites along the main stream (Fig. 2) may be mainly affected by dilution effect.

## DISCUSSION

### Concentrations of heavy metals in river water

The concentration of dissolved heavy metals are compared with the limited values for drinking water guidelines of *China EPA (2006)*, *World Health Organization (WHO) (2006)*,

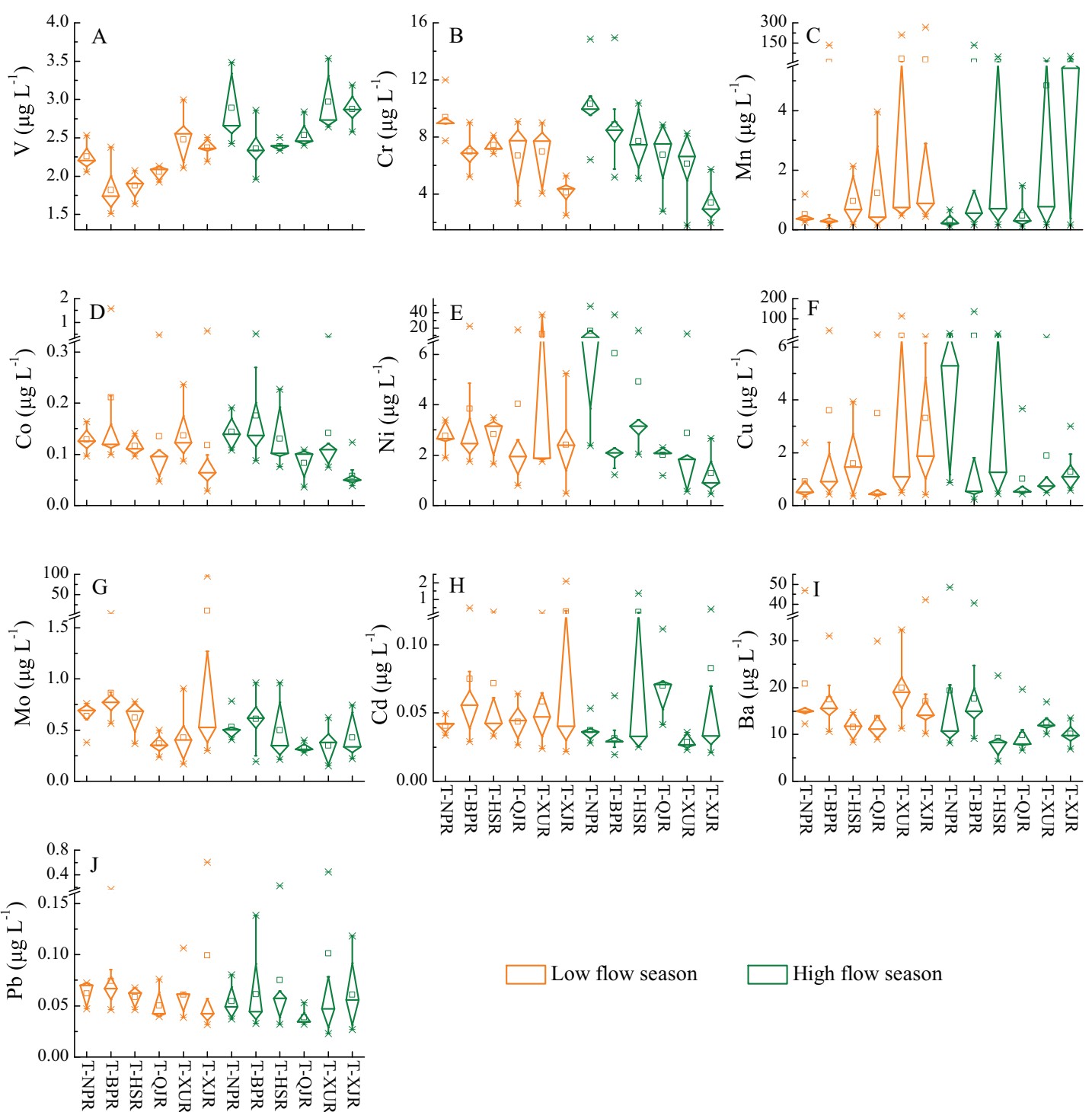

**Figure 3 Distribution of 10 heavy metals in tributaries from Zhujiang River.** T-NPR, tributaries of Nanpanjiang River; T-BPR, tributaries of Beipanjiang River; T-HSR, tributaries of Hongshuihe River; T-QJR, tributaries of Qianjiang River; T-XUJ, tributaries of Xunjiang River; T-XJR, tributaries of Xijiang River. (A) V concentrations. (B) Cr concentrations. (C) Mn concentrations. (D) Co concentrations. (E) Ni concentrations. (F) Cu concentrations. (G) Mo concentrations. (H) Cd concentrations. (I) Ba concentrations. (J) Pb concentrations.

*United States Environmental Protection Agency (EPA) (2003)* (Table 1) and the surface water standard of *China EPA (2002)* (Table S2). All the concentrations (median) of heavy metals are within the limited values of three drinking water guidelines (except V, without limited values in any guidelines), and the Cr, Cu, Cd, and Pb concentrations (median) are of Grade I of surface water standard, which indicate that pretty clean water without heavy metals pollution in Zhujiang River. In addition, the maximum concentrations of Mn and Ni are lower than the limited values of WHO drinking water guideline but are approximately two times higher than the Chinese drinking water guideline, and the maximum concentrations of Mo in low flow season is higher than the Chinese drinking water guideline. These heavy metals with maximum concentrations exceed guideline values can be defined as pollutants, which could attribute to the relatively high anthropogenic input from local sampling sites. For instance, although the Cu concentrations at all sampling sites are well below the guideline values, but the Cu concentrations at the two sampling sites (site 1 and 2, close to the Zhanyi county) of NPR reach along the main stream are significantly higher than that of other sites (Fig. 2), which shows the significant impact of the anthropogenic input from the Zhanyi county (e.g., urban sewage discharge). Compare with the background values of the source area of Yangtze River (Table 1) (*Zhang & Zhou, 1992*), the concentrations (median) of V, Cr, Ni, Cu, and Cd in water within the Zhujiang River are significantly elevated, while Mn, Co, Mo, and Pb are lower than the source area of the Yangtze River. In low flow season, V, Cr, Ni, Cu, and Cd are 9.2, 26.4, 13.3, 1.4, and 2.9 times higher than the river water background of the Yangtze River. In high flow season, V, Cr, Ni, Cu, and Cd are 11.2, 28.9, 11.5, 1.2, and 2.2 times higher than the background values.

On a global scale, we compare our river samples with the data of other rivers both from China and other counties or regions worldwide (*Gaillardet, Viers & Dupré, 2014*) (Table S3). V, Cr, and Ni, concentrations of our study are higher than those worldwide rivers (except Seine River, a river severely affected by human activities in France). The Co and Mo concentrations are comparable to the worldwide rivers average concentration, but the Co concentration is much higher than that in Yellow River of China. Moreover, the Cu, Cd, Ba, and Pb concentrations are slightly lower than the world river average, while the Mn concentration is much lower than world river average.

## Statistical analysis and source identification of heavy metals

### Correlation analysis

The correlation matrix is useful for exploring associations between variables via giving the overall coherence of the dataset (*Chen et al., 2007*). A correlation matrix is employed to distinguish correlations between the 10 heavy metals in the two sampling periods (Table 2). Strong positive correlations ($p < 0.01$) are only observed between Co and Mn (0.591), Pb and Mn (0.673), Cu and Ni (0.650) in low flow season, and the positive correlations between Pb and Co (0.284) are also obtained in same season. In contrast, the strong positive correlations ($p < 0.01$) between Co and Cr (0.477), Co and Mn (0.520), Cu and Ni (0.538) are observed in high flow season (Table 2). The heavy metals with high correlation coefficients in the water body could have similar sources, migration
**Table 2 Pearson correlation matrix of heavy metals and physicochemical parameters (pH, EC, DO, and T) in the Zhujiang River, China.**

|     | V | Cr | Mn | Co | Ni | Cu | Mo | Cd | Ba | Pb | pH | EC | DO | T |
|-----|---|----|----|----|----|----|----|----|----|----|----|----|----|---|
| V | 1 | −0.176 | −0.144 | −0.254* | 0.126 | −0.146 | 0.085 | −0.069 | 0.124 | −0.012 | −0.462** | −0.187 | −0.625** | 0.316** |
| Cr | −0.017 | 1 | −0.241* | 0.477** | 0.308** | 0.081 | 0.209 | −0.094 | 0.421** | 0.006 | 0.563** | 0.852** | 0.117 | −0.552** |
| Mn | 0.021 | −0.290** | 1 | 0.520** | −0.059 | 0.077 | 0.215 | 0.227* | −0.119 | 0.079 | −0.129 | −0.081 | −0.028 | −0.076 |
| Co | −0.177 | 0.014 | 0.591** | 1 | 0.122 | 0.046 | 0.215 | 0.040 | 0.252* | 0.006 | 0.280* | 0.481** | 0.082 | −0.372** |
| Ni | −0.050 | −0.148 | 0.312** | 0.041 | 1 | 0.538** | 0.178 | −0.040 | 0.303** | 0.012 | 0.235* | 0.323** | 0.029 | −0.316** |
| Cu | −0.025 | −0.176 | −0.005 | −0.009 | 0.650** | 1 | 0.103 | −0.055 | 0.086 | 0.050 | 0.206 | 0.182 | 0.082 | −0.328** |
| Mo | 0.034 | −0.172 | −0.028 | −0.063 | −0.048 | 0.017 | 1 | 0.079 | 0.274* | −0.006 | 0.493** | 0.498** | 0.243* | −0.190 |
| Cd | 0.143 | −0.159 | −0.028 | −0.036 | −0.029 | −0.025 | 0.114 | 1 | −0.117 | 0.279* | 0.018 | 0.084 | −0.001 | 0.075 |
| Ba | 0.287** | 0.219* | 0.059 | −0.011 | 0.155 | 0.021 | −0.016 | 0.008 | 1 | −0.035 | 0.216 | 0.446** | 0.001 | −0.216 |
| Pb | 0.008 | −0.130 | 0.673** | 0.284* | −0.001 | −0.008 | −0.036 | −0.044 | −0.050 | 1 | 0.049 | 0.018 | 0.173 | 0.082 |
| pH | −0.494** | 0.594** | −0.241* | 0.086 | −0.280* | −0.319** | −0.032 | −0.148 | 0.123 | −0.089 | 1 | 0.702** | 0.668** | −0.400** |
| EC | −0.360** | 0.773** | −0.237* | 0.202 | −0.100 | −0.130 | −0.154 | −0.084 | 0.203 | −0.111 | 0.771** | 1 | 0.215 | −0.660** |
| DO | −0.125 | −0.186 | 0.008 | 0.070 | −0.015 | −0.065 | 0.166 | 0.071 | −0.013 | −0.010 | 0.068 | 0.003 | 1 | −0.040 |
| T | 0.202 | −0.326** | 0.053 | −0.235* | 0.136 | 0.154 | 0.016 | −0.029 | −0.147 | 0.057 | −0.567** | −0.532** | −0.108 | 1 |

Notes:
* Correlation is significant at the 0.05 level (two-tailed).
** Correlation is significant at the 0.01 level (two-tailed).
Bold italics: correlation coefficients in high flow season; Normal font: correlation coefficients in low flow season.

processes and hydrochemical behavior in the study area (*Wang et al., 2017*). Therefore, Mn, Co, and Pb could origin from similar sources and input to the water body after similar chemical processes in low flow season, while Co and Cr could have similar sources in high flow season. Also, the consistently strong positive correlations between Cu and Ni in the two periods indicate that the origins and migration of these two metals are remarkably similar. However, the weak positive correlations, different degrees of negative correlation or without significant correlation between each pair of remaining heavy metals are also observed (Table 2), which indicate that there is strong spatial and temporal heterogeneity of the sources of these heavy metals.

### Principal component analysis

Principal component analysis is carried out for heavy metal concentrations along the main channel of Zhujiang River to explore metal associations and their possible origins, three principal components (PC, eigenvalues > 1) are extracted in our study, including the eigenvalues, variance and communalities, are listed in Table 3. The PC 1 explain 22.59% of total variance and predominantly include V, Cr, Co, and Ba; the PC 2 explain 20.91% of total variance with significant loadings of Mn, Ni, and Cu; the PC 3 explain 18.23% of variance which is mainly contributed by Mo, Cd, and Pb. These three PCs totally account for 61.74% of the total variance, and are presented in a three-dimensional space, as shown in Fig. 4. Our results of overall PCs loadings (61.74%) are relatively lower than other studies, that is, PCs loadings for 14 heavy metals and 13 metals are 86.36 % and 79.31% (*Meng et al., 2016*; *Wang et al., 2017*), respectively. Previous studies have performed PCA for heavy metals in numerous of river system and attained different results (*Li & Zhang, 2010*; *Wang et al., 2017*; *Xiao, Jin & Wang, 2014*), we ascribe these

**Table 3 Varimax rotated component matrix for dissolved heavy metals along the main channel of Zhujiang River (the significance of KMO and Bartlett's sphericity test is <0.001).**

| Eigenvalues | 2.26 | 2.09 | 1.82 | Communalities |
|---|---|---|---|---|
| Variance (%) | 22.59 | 20.91 | 18.23 | |
| Cumulative (%) | 22.59 | 43.50 | 61.74 | |
| Variable | PC1 | PC 2 | PC 3 | |
| V | 0.18 | −0.22 | −0.73 | 0.60 |
| Cr | 0.85 | −0.02 | −0.12 | 0.74 |
| Mn | −0.20 | 0.81 | −0.01 | 0.71 |
| Co | 0.84 | 0.04 | 0.12 | 0.72 |
| Ni | 0.10 | 0.67 | 0.03 | 0.46 |
| Cu | −0.08 | 0.89 | −0.04 | 0.80 |
| Mo | 0.32 | −0.29 | 0.47 | 0.40 |
| Cd | −0.03 | −0.14 | 0.71 | 0.53 |
| Ba | 0.79 | −0.15 | −0.04 | 0.66 |
| Pb | 0.05 | 0.05 | 0.74 | 0.55 |

**Notes:**
Extraction method: Principal component analysis.
Rotation method: Varimax with Kaiser normalization.

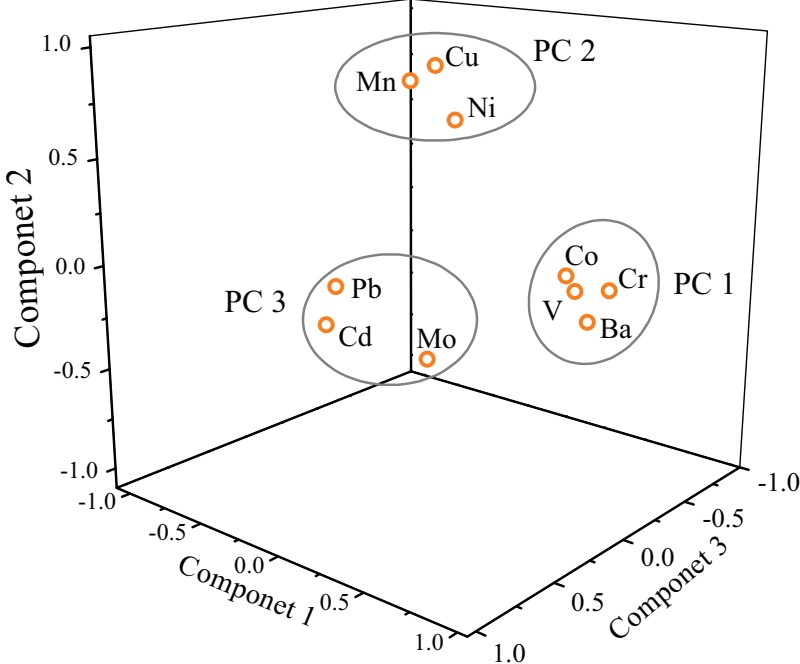

**Figure 4 3D plot of scores obtained from PCA results for dissolved heavy metals along the main channel of Zhujiang River.**

variations to the different fluvial environments (i.e., watershed area, discharge, and land use) and different water variables.

In general, the Cr, Co, Ba, and Mo are from natural sources of rock weathering and subsequent pedogenesis (*Li & Zhang, 2010*), and V is greatly impacted by anthropogenic activities such as mining and agricultural processes (*Li et al., 2008*). Considering the high

concentrations of V and Cr (much higher than the background value of the Yangtze River, Table 1), and the most of elements in PC1 (V, Cr, Co, Ba) are lithophile elements (*Krishna, Satyanarayanan & Govil, 2009*), hence we attribute this component (PC 1) to the mixed sources of geologic and anthropogenic origins in the basin. The high concentrations of Cu and Ni in PC 2 are observed in Zhujiang River (compared to the background value), and Cu is used as markers of metal industries (*Li & Zhang, 2010*), while Ni is the common pollutant discharge from the electroplating industry and metal smelting. In combination with the negative loadings or weak positive loadings of Cu and Ni in PC 1 and the positive correlations between Cu and Ni (Table 2), the PC 2 with positive loadings on Cu (0.89) and Ni (0.67) can be attribute to anthropogenic origins in the basin. In addition, although Cd and Pb are major pollutants emitted from industrial wastes and automobile exhausts (*Krishna, Satyanarayanan & Govil, 2009*; *Pekey, Karakaş & Bakogˇlu, 2004*), but the concentrations of these two heavy metals and Mo are not high in the Zhujiang River, even lower than the background value of the Yangtze River. Therefore, Cd, Pb, and Mo in PC 3 may be greatly contributed by natural sources.

In spite of the results of PCA that suggest that some heavy metals (V, Cr, Ni, and Cu) are possibly caused by the anthropogenic inputs or the mixed sources of geologic and anthropogenic origins, the rest of the heavy metals with low concentrations below the background value are controlled by the natural sources or assuaged by the varying landscape setting and the relatively weak rock weathering processes. Compared to the polluted rivers such as Huaihe River (Cu (28.61 $\mu$g L$^{-1}$), Pb (97.83 $\mu$g L$^{-1}$), Ni (16.00 $\mu$g L$^{-1}$), Cr (19.7 $\mu$g L$^{-1}$)) (*Wang et al., 2017*), the concentration level of heavy metals in the Zhujiang River are fairly low (Table 1). Therefore, we hold that dilution has a great impact on the anthropogenic heavy metals (V, Cr, Ni, and Cu) within such a large catchment area. The dissolved heavy metals in Zhujiang River are mainly presented a natural sources signature.

## Water quality and health risk in river water

The allowable limit of pH for drinking water ranges from 6.5 to 8.5 (Table 1). The pH values in 95.7% of the water samples are conformed to this regulation in our study. Based on PCA results, the weights of each heavy metal ($w_i$) are obtained and summarized in Table 4. Therefore, the calculated WQI values using Eq. (1) are shown in Table S1. In this study, the WQI values ranges from 1.5 to 43.9 and 1.3 to 27.9 for water samples during the low and high flow season, respectively. The calculated WQI values based on the median concentrations of heavy metals in the two seasons are 3.3 and 3.2. Only two sites (site 61 and 74) are evaluated as a relatively high WQI value (43.9 and 33.5) in low flow season. All the water samples can be defined as excellent water quality (the calculated WQI values are less than 50), indicative of that the natural water within the Zhujiang River drainage area is suitable for drinking, at least from the perspective of heavy metal pollution.

The HQ and HI values of the heavy metals via ingestion and dermal pathways for adults and children are calculated with the median concentrations based on Eqs. (2)–(6), respectively. As shown in Table 5, for both adults and children, the HQ$_{ingestion}$,

**Table 4 Weights for the 10 dissolved heavy metals in the water samples from the Zhujiang River.**

| PC | Eigenvalue | Relative eigenvalue | Variable | Loading value | Relative loading value on same PC | Weight (relative eigenvalue × relative loading value) |
|---|---|---|---|---|---|---|
| 1 | 2.26 | 0.37 | V | 0.18 | 0.07 | 0.02 |
| | | | Cr | 0.85 | 0.32 | 0.12 |
| | | | Co | 0.84 | 0.32 | 0.12 |
| | | | Ba | 0.79 | 0.30 | 0.11 |
| | | | Total | 2.66 | 1.00 | |
| 2 | 2.09 | 0.34 | Mn | 0.81 | 0.34 | 0.12 |
| | | | Ni | 0.67 | 0.28 | 0.10 |
| | | | Cu | 0.89 | 0.37 | 0.13 |
| | | | Total | 2.38 | 1.00 | |
| 3 | 1.82 | 0.30 | Mo | 0.47 | 0.24 | 0.07 |
| | 6.17 | | Cd | 0.71 | 0.37 | 0.11 |
| | | | Pb | 0.74 | 0.38 | 0.11 |
| | | | Total | 1.91 | 1.00 | 1.00 |

$HQ_{dermal}$, and HIs for all heavy metals are smaller than 1, indicate that the metals investigated in the Zhujiang River basin are all below the hazard level (through oral ingestion and dermal absorption), and the health effects of these heavy metals are very limited. It should be noted that the $HQ_{ingestion}$ and $HQ_{dermal}$ values for children are relatively higher than adults, suggest that children are more risky under the heavy metals exposure. Moreover, at some sampling points with the maximal heavy metal concentrations, the HI of Mo (0.539 and 0.826 for adults and children, respectively; site 68) and Mn (0.573 for children; site 74) are relatively close to 1 in low flow season. Consequently, we conclude that Mo and Mn could be a potential non-carcinogenic risk to human health in some sites, especially in low flow season.

In spite of the risk, assessment indicates that widespread heavy metal pollutants are not in the Zhujiang River basin, which is similar to the results from WQI. Several studies have reported the adverse effects of heavy metals, such as the toxicity of Pb on the nervous and endocrine systems in the human body (*Fang et al., 2014*); the adverse effects of Cd in the renal system and bone injuries (*Bertin & Averbeck, 2006*); the effects of Mo on reproduction and foetal development (*Vyskocil & Viau, 1999*). Therefore, special measures should be taken to prevent the input of heavy metals into the Zhujiang River for the protection of its excellent water quality, and to provide a better water resources guarantees for socio-economic development within the whole basin.

## Heavy metals export budget of Zhujiang River to the South China Sea

By taking the concentrations of the dissolved heavy metals in low flow season (October–March of the following year) and high flow season (April–September), and the discharge at the mouth of the Zhujiang River of the two seasons (River and Sediment Bulletin of China, http://www.mwr.gov.cn/sj/tjgb/zghlnsgb/), river fluxes of individual of heavy metal to the SCS are estimated that range from 8.6 (Pb) to 3,713.5 (Ba) tons during
Table 5 Hazard quotient (HQ), dermal permeability coefficient ($K_p$), and reference dose (RfD) for dissolved heavy metals in the Zhujiang River.

| | $HQ_{ingestion}$ | | $HQ_{dermal}$ | | $HI = \Sigma HQs$ | | $K_p{}^{a}$ cm h$^{-1}$ | $RfD_{ingestion}{}^{b,c}$ µg kg$^{-1}$day$^{-1}$ | $RfD_{dermal}{}^{b,c}$ µg kg$^{-1}$day$^{-1}$ |
|---|---|---|---|---|---|---|---|---|---|
| | Adult | Children | Adult | Children | Adult | Children | | | |
| Low flow season | | | | | | | | | |
| V | $5.83 \times 10^{-2}$ | $8.70 \times 10^{-2}$ | $6.08 \times 10^{-2}$ | $1.80 \times 10^{-1}$ | $1.19 \times 10^{-1}$ | $2.67 \times 10^{-1}$ | $2.00 \times 10^{-3}$ | 1 | 0.01 |
| Cr | $6.26 \times 10^{-2}$ | $9.34 \times 10^{-2}$ | $1.31 \times 10^{-2}$ | $3.85 \times 10^{-2}$ | $7.56 \times 10^{-2}$ | $1.32 \times 10^{-1}$ | $1.00 \times 10^{-3}$ | 3 | 0.075 |
| Mn | $5.10 \times 10^{-4}$ | $7.62 \times 10^{-4}$ | $6.66 \times 10^{-5}$ | $1.96 \times 10^{-4}$ | $5.77 \times 10^{-4}$ | $9.58 \times 10^{-4}$ | $1.00 \times 10^{-3}$ | 24 | 0.96 |
| Co | $1.02 \times 10^{-2}$ | $1.53 \times 10^{-2}$ | $1.07 \times 10^{-4}$ | $3.15 \times 10^{-4}$ | $1.03 \times 10^{-2}$ | $1.56 \times 10^{-2}$ | $4.00 \times 10^{-4}$ | 0.3 | 0.06 |
| Ni | $3.27 \times 10^{-3}$ | $4.89 \times 10^{-3}$ | $8.55 \times 10^{-5}$ | $2.52 \times 10^{-4}$ | $3.36 \times 10^{-3}$ | $5.14 \times 10^{-3}$ | $2.00 \times 10^{-4}$ | 20 | 0.8 |
| Cu | $6.16 \times 10^{-4}$ | $9.19 \times 10^{-4}$ | $1.61 \times 10^{-5}$ | $4.74 \times 10^{-5}$ | $6.32 \times 10^{-4}$ | $9.67 \times 10^{-4}$ | $1.00 \times 10^{-3}$ | 40 | 8 |
| Mo | $3.45 \times 10^{-3}$ | $5.15 \times 10^{-3}$ | $9.47 \times 10^{-5}$ | $2.79 \times 10^{-4}$ | $3.54 \times 10^{-3}$ | $5.43 \times 10^{-3}$ | $2.00 \times 10^{-3}$ | 5 | 1.9 |
| Cd | $2.40 \times 10^{-3}$ | $3.58 \times 10^{-3}$ | $2.51 \times 10^{-4}$ | $7.39 \times 10^{-4}$ | $2.65 \times 10^{-3}$ | $4.32 \times 10^{-3}$ | $1.00 \times 10^{-3}$ | 0.5 | 0.025 |
| Ba | $2.02 \times 10^{-3}$ | $3.01 \times 10^{-3}$ | $1.50 \times 10^{-4}$ | $4.44 \times 10^{-4}$ | $2.17 \times 10^{-3}$ | $3.46 \times 10^{-3}$ | $1.00 \times 10^{-3}$ | 200 | 14 |
| Pb | $1.10 \times 10^{-3}$ | $1.65 \times 10^{-3}$ | $1.92 \times 10^{-6}$ | $5.66 \times 10^{-6}$ | $1.10 \times 10^{-3}$ | $1.65 \times 10^{-3}$ | $1.00 \times 10^{-4}$ | 1.4 | 0.42 |
| High flow season | | | | | | | | | |
| V | $7.09 \times 10^{-2}$ | $1.06 \times 10^{-1}$ | $7.40 \times 10^{-2}$ | $2.18 \times 10^{-1}$ | $1.45 \times 10^{-1}$ | $3.24 \times 10^{-1}$ | $2.00 \times 10^{-3}$ | 1 | 0.01 |
| Cr | $6.87 \times 10^{-2}$ | $1.03 \times 10^{-1}$ | $1.43 \times 10^{-2}$ | $4.23 \times 10^{-2}$ | $8.30 \times 10^{-2}$ | $1.45 \times 10^{-1}$ | $1.00 \times 10^{-3}$ | 3 | 0.075 |
| Mn | $4.54 \times 10^{-4}$ | $6.78 \times 10^{-4}$ | $5.93 \times 10^{-5}$ | $1.75 \times 10^{-4}$ | $5.13 \times 10^{-4}$ | $8.53 \times 10^{-4}$ | $1.00 \times 10^{-3}$ | 24 | 0.96 |
| Co | $9.86 \times 10^{-3}$ | $1.47 \times 10^{-2}$ | $1.03 \times 10^{-4}$ | $3.04 \times 10^{-4}$ | $9.96 \times 10^{-3}$ | $1.50 \times 10^{-2}$ | $4.00 \times 10^{-4}$ | 0.3 | 0.06 |
| Ni | $2.84 \times 10^{-3}$ | $4.25 \times 10^{-3}$ | $7.43 \times 10^{-5}$ | $2.19 \times 10^{-4}$ | $2.92 \times 10^{-3}$ | $4.47 \times 10^{-3}$ | $2.00 \times 10^{-4}$ | 20 | 0.8 |
| Cu | $5.27 \times 10^{-4}$ | $7.87 \times 10^{-4}$ | $1.38 \times 10^{-5}$ | $4.06 \times 10^{-5}$ | $5.41 \times 10^{-4}$ | $8.28 \times 10^{-4}$ | $1.00 \times 10^{-3}$ | 40 | 8 |
| Mo | $2.41 \times 10^{-3}$ | $3.60 \times 10^{-3}$ | $6.63 \times 10^{-5}$ | $1.95 \times 10^{-4}$ | $2.48 \times 10^{-3}$ | $3.80 \times 10^{-3}$ | $2.00 \times 10^{-3}$ | 5 | 1.9 |
| Cd | $1.80 \times 10^{-3}$ | $2.69 \times 10^{-3}$ | $1.88 \times 10^{-4}$ | $5.55 \times 10^{-4}$ | $1.99 \times 10^{-3}$ | $3.24 \times 10^{-3}$ | $1.00 \times 10^{-3}$ | 0.5 | 0.025 |
| Ba | $1.71 \times 10^{-3}$ | $2.56 \times 10^{-3}$ | $1.28 \times 10^{-4}$ | $3.77 \times 10^{-4}$ | $1.84 \times 10^{-3}$ | $2.93 \times 10^{-3}$ | $1.00 \times 10^{-3}$ | 200 | 14 |
| Pb | $8.50 \times 10^{-4}$ | $1.27 \times 10^{-3}$ | $1.48 \times 10^{-6}$ | $4.37 \times 10^{-6}$ | $8.52 \times 10^{-4}$ | $1.27 \times 10^{-3}$ | $1.00 \times 10^{-4}$ | 1.4 | 0.42 |
| Low flow season, calculated by the maximal concentrations | | | | | | | | | |
| Mn | $3.05 \times 10^{-1}$ | $4.56 \times 10^{-1}$ | $3.98 \times 10^{-2}$ | $1.17 \times 10^{-1}$ | $3.45 \times 10^{-1}$ | $5.73 \times 10^{-1}$ | $1.00 \times 10^{-3}$ | 24 | 0.96 |
| Mo | $5.25 \times 10^{-1}$ | $7.84 \times 10^{-1}$ | $1.44 \times 10^{-2}$ | $4.25 \times 10^{-2}$ | $5.39 \times 10^{-1}$ | $8.26 \times 10^{-1}$ | $2.00 \times 10^{-3}$ | 5 | 1.9 |

Notes:
[a] US EPA (2004).
[b] Wang et al. (2017).
[c] Wu et al. (2009).

the hydrologic year of 2014–2015 (Table 6). It should be noted that 80%, 76%, 25%, 71%, 58%, 63%, 70%, 80%, 74%, and 63% of the V, Cr, Mn, Co, Ni, Cu, Mo, Cd, Ba, and Pb flux produced in the high flow season, which is mainly controlled by the high discharge ($1.72 \times 10^{11}$ m$^3$, account for 74.1% discharge of whole hydrologic year) in this season. Table 6 also presented the estimated heavy metal fluxes of Zhujiang River in 2002, which are calculated by the heavy metal concentrations from previous study (Ouyang et al., 2004) and the discharge in the corresponding year. Most of the estimated heavy metal fluxes are decreased from 2002 to 2015 for the Zhujiang River (except V) (Table 6). Thus, the environmental policy in last decade might reduce the anthropogenic heavy metal inputs to the Zhujiang River in China. Compared with literature values (Gaillardet, Viers & Dupré, 2014), the proportion of the annual estimated export fluxes of V, Cr, Mn, Co,

**Table 6 Export fluxes estimation of heavy metals in two seasons (t), annual export flux of Zhujiang River (t year$^{-1}$) to the South China Sea, and the world riverine flux (kt year$^{-1}$).**

| Flux | V | Cr | Mn | Co | Ni | Cu | Mo | Cd | Ba | Pb |
|---|---|---|---|---|---|---|---|---|---|---|
| Low flow season (This study) | 148.3 | 380.0 | 79.5 | 5.7 | 210.1 | 65.1 | 36.1 | 2.3 | 968.1 | 3.1 |
| High flow season (This study) | 587.3 | 1,181.1 | 26.4 | 13.8 | 288.1 | 108.9 | 82.8 | 9.1 | 2,745.4 | 5.4 |
| Zhujiang River (This study) | 735.6 | 1,561.1 | 105.9 | 19.5 | 498.2 | 173.9 | 118.9 | 11.3 | 3,713.5 | 8.6 |
| Zhujiang River (2002)[a] | 257.3 | 2,860.9 | 11,178.5 | 35.3 | 1,904.8 | 1,455.6 | 339.3 | 12.6 | 6,007.6 | 99.0 |
| World riverine flux[b] | 27.0 | 26.0 | 1,270.0 | 5.5 | 30.0 | 55.0 | 16.0 | 3.0 | 860.2 | 3.0 |

Notes:
[a] The data for fluxes calculation is from *Ouyang et al. (2004)*.
[b] *Gaillardet, Viers & Dupré (2014)*.

Ni, Cu, Mo, Cd, Ba, and Pb of Zhujiang River in the global rivers are 2.7%, 6.0%, 0.01%, 0.4%, 1.7%, 0.3%, 0.7%, 0.4%, 0.4%, and 0.3%, respectively. Based on the annual river discharge of $3.74 \times 10^{13}$ m$^3$ year$^{-1}$ for global rivers (*Xu & Han, 2009*), the annual discharge of the Zhujiang River in the total annual discharge of the world's rivers is estimated as 0.6%. The proportion of V, Cr, Ni, and Mo export fluxes of Zhujiang River in global rivers exceeds 0.6%, which suggests that the contribution of V, Cr, Ni, and Mo of Zhujiang River export to the marine system are higher than the world average level.

In this study, the concentrations of heavy metals in the estuary of the Zhujiang River present different degrees in the two seasons (Fig. 2). Given that most of the heavy metals export occurred in the rainy season, and there may be large uncertainty in estimation of heavy metal fluxes due to the limited temporal sampling interval in the Zhujiang River, particularly in the high flow season with a conspicuous variation of heavy metal concentrations after a storm event, our data from one sole sampling campaign may be incomplete. Therefore, high-frequency sampling (monthly, weekly even daily sampling) is necessary and is vital significance to precisely quantify heavy metal annual budget from the terrestrial rivers to the ocean.

## CONCLUSIONS

In conclusion, the dissolved heavy metals in the Zhujiang River water exhibit little variation on temporal scale, but vary with a significant spatial heterogeneity. Ba and Cr are consistently the most abundant metals in the two sampling periods. All the concentrations of heavy metals are within the limited values for Chinese drinking water guidelines except for few specific sites, and the water quality and health risk assessment reveal that the metals investigated in Zhujiang River are below the hazard level with a low risk. PCA results conclude that the dissolved heavy metals in Zhujiang River are mainly presented a natural sources signature, while the anthropogenic heavy metals (V, Cr, Ni, and Cu) are greatly impacted by dilution effect. Moreover, the estimation of export flux suggest that the contribution of V, Cr, Ni, and Mo of Zhujiang River export to the marine system are higher than the world average level. Overall, the water quality is pretty good in the Zhujiang River, but corresponding measures should also be taken to provide a better water resources guarantee for socio-economic development within the whole basin in the future.

## ACKNOWLEDGEMENTS

The authors thank Dr. Danyang Zhang from China University of Geosciences (Beijing) for sample analyses.

### Funding

This work was supported by the National Natural Science Foundation of China (No. 41325010). The funders had no role in study design, data collection and analysis, decision to publish, or preparation of the manuscript.

### Grant Disclosure

The following grant information was disclosed by the authors:
National Natural Science Foundation of China: 41325010.

### Competing Interests

The authors declare that they have no competing interests.

### Author Contributions

- Jie Zeng conceived and designed the experiments, analyzed the data, prepared figures and/or tables, authored or reviewed drafts of the paper, approved the final draft.
- Guilin Han conceived and designed the experiments, performed the experiments, analyzed the data, contributed reagents/materials/analysis tools, authored or reviewed drafts of the paper, approved the final draft.
- Qixin Wu authored or reviewed drafts of the paper, approved the final draft.
- Yang Tang performed the experiments, contributed reagents/materials/analysis tools, approved the final draft.

### Data Availability

The raw measurements are available in Files S1 and S4.

### Supplemental Information

Supplemental information for this article can be found online at http://dx.doi.org/10.7717/peerj.6578#supplemental-information.

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
