# Peer review of "Geochemical characteristics of dissolved heavy metals in Zhujiang River, Southwest China: spatial-temporal distribution, source, export flux estimation, and a water quality assessment"

_PeerJ, doi:10.7717/peerj.6578_

## Round 0.1 · original submission · Major Revisions

I originally chose "reject" because it seems that additional experiments and analyses are needed to support the "controlling factor" and "export flux" claims that are in the title. DOC and Fe measurements are lacking which might support the controlling factor hypothesis and more robust temporal sampling would support the export flux hypothesis. Because additional experiments/analyses are needed, the manuscript would likely change significantly, and thus should be treated as a new manuscript.

However, the authors have appealed and after reviewing the appeal, I feel they may be able to address the previous comments. Therefore, I am issuing this Revision decision to give the authors a chance to respond to the reviews and submit a revised version which addresses the concerns.

· Appeal

Appeal

My name is Jie Zeng, and I am one of the authors of the manuscript (PeerJ submission) - Geochemical characteristics of dissolved heavy metals in Zhujiang River, Southwest China: Spatial-temporal distribution, controlling factor, export flux, and a water quality assessment –.

Thank you for your work on our manuscript, and we also thank the reviews of two anonymous reviewers and Dr. Oleg Pokrovsky.

I am writing to ask you whether our manuscript could be submitted again as a revised version, instead of a new manuscript.

According to your decision, our manuscript has been rejected based on the need for additional experiments and analyses (DOC and iron measurements and a more robust temporal sampling). Actually, (1) we measured the DOC content of some samples (upstream of the Zhujiang river) after sampling in 2014-2015. Unfortunately, since the DOC Analyzer instrument failure, we did not measure the DOC content of all samples in the whole basin timely. However, it is unreasonable to do the additional experiments of DOC now, because of the DOC content in the sample may change significantly over the past four years (such as degradate to inorganic carbon). (2) Iron is not discussed in our manuscript mainly based on the relevant content in health risk assessment, because iron is not an extremely toxic element compared with other heavy metals (such as Cr, Cd, and Pb) in our manuscript. (3) Finally, for the more robust time sampling. We have made a rough "estimation" of the export fluxes of heavy metals based on the existing samples and data. But the expression "quantified" is used in our manuscript, which is really inappropriate. In fact, when we discuss the export fluxes, we also mentioned the deficiency of our estimation, and referred to the importance of high-frequency sampling for accurately quantified estimation. It would also be an important direction of our future work but not in this manuscript.

We would carefully revise or reply to other comments of three reviewers. Based on this, we would like to ask for the chance to submit our manuscript as a revised version again, instead of a new manuscript.


· · Academic Editor

Reject

I am choosing to reject your manuscript based on the need for additional experiments and analyses (DOC and iron measurements and a more robust temporal sampling) to support your "controlling factor" and "export flux" hypotheses. Because additional experiments/analyses are needed, the manuscript would likely change significantly, and thus should be treated as a new manuscript. I hope the additional comments raised by the reviewers, particularly reviewer 2, will help improve your research.

Reviewer 1 ·

Basic reporting

no comment

Experimental design

no comment

Validity of the findings

no comment

Additional comments

The introduction ofarticle sufficiently demonstrates the need for research. The research question is clearly defined, relevant and of scientific significance.
The methods are described in detail and with all the necessary information.
The article is written in technically correct language. The structure of the article corresponds to the format recommended by the authors.
The data are reliable, statistically reliable, provided in the right quantity.
The conclusion is well formulated, linked to the original research question, and limited to confirmatory results. Overall, the research is very interesting, important, and it is certainly worthy of publication in the journal. This study has potential to be an important paper in this field. The article corresponds to the standards of the journal, it is a continuation of previously published articles and adds an useful information to the knowledge base.
There are a few issues that I think should be considered prior to publication. Most of these are relatively minor.
Line 87: it seems incredible - average annual precipitation ranges from 1200 to 2200 m.
Line 104: Link to Table S1, but the table is missing.
Line 382: Link to Table S1, but the table is missing.
On the graphs, the name of the axes in Figure 2 and in Figure 3 should be brought to a generally accepted standard. This would correctly correct the notation µg / L to µg L-1.
In Table 5, I recommend changing the format of the data presented. This format (1.47E-02) is very difficult to read.

Annotated reviews are not available for download in order to protect the identity of reviewers who chose to remain anonymous.

Reviewer 2 ·

Basic reporting

See below

Experimental design

See below

Validity of the findings

See below

Additional comments

The manuscript has a nice comprehensive dataset presenting a large spatial-temporal distribution of metals in the Zhujiang River. And the work on calculating WQI and health risk assessment is very helpful for readers. More importantly, the authors did find a significant heterogeneity in season and space. Therefore PCA and export fluxes are also important and meaningful for scientists in the community. These are the major strength of the MS.

Major concerns here is as following:

1) The authors made a statement that the water in Zhujiang is suitable for drinking with a low risk based on all hazard index <1. Here obviously factors influencing water quality is more than several metals in MS, and generally high E. coli or some toxic metals like As commonly occur at some sites. As the work done here never cover those more important parameters, I would suggest the authors narrowing down the statement. For example, the metals measured in this study are all below the hazard level (which is ~~~) if this is true.

2) PCA led to a conclusion “dissolved heavy metals in Zhujiang River are mainly derived from natural sources 35 controlled by rock weathering, while the contribution of anthropogenic source to heavy metals in 36 river water is relatively weak”. I agree with the point, and however, I didn’t see the logic between PCA and the conclusion. Instead I will suggest that the author consider list which metal(s) are good indicators of rock weathering, (like Ba), which are good for anthropogenic source (e.g., Pb, Cu). And carefully discuss their pattern, and make a more solid logic here. Previously several groups of researchers have done mineral composition work in the watershed, for example, Jing Zhang’s group. Also there is a huge amount of published papers which could be easily assessed online.

3) Export budget shows with higher fluxes for several metals including V, Cr, Ni, Ni, Mo. Considering the Zhujiang is 13th largest river in the world. It is not surprising, that many metals have higher fluxes than the world average. I suggest that the author go further and think add more calculations, for example, compare the differences of two seasons.

4) Another big issue is that several metals are easily contaminated during sampling and processing. One example is Pb, which could be easily contaminated due to tiny dust in atmosphere. Although the river water usually have high levels of these metals, careful sampling and processing is still important for high quality. In the dataset, I see a few spots, for example, one point of Pb at XJR with relatively high level. Most of the data are in good quality and make sense, however please double check those discrete points with extremely high values.

5) Another issue is the high levels of Cu in NPR in two spots. By the way, that level of Cu (~70ug/L) is not safe any more based on drinking standard in China. I would again suggest the authors refrasing sentences within the MS accordingly. These data should clearly suggest of anthropogenic sources, and so further anthropogenic source must be identified or suggested, although only a few spots that I think these measurements are correct.

·

Basic reporting

The manuscript by Jie Zeng et al. addresses environmentally important question and the general topics is pertinent to the profile of the journal. However, the study has little added value as neither the controlling factors nor the fluxes of metals could be determined. The title therefore does not reflect the content of the paper.

First, the choice of metals is arbitrary and incomplete, despite the fact that ICP MS was used. The authors did not measure Fe, Al, Zn, Sr.
The DOC and Fe are widely recognized as main carriers of trace metals in river waters, in the form of organic an organo-mineral colloids. I do not request characterizing those colloids, but neither Fe nor DOC was analyzed in this study.
Finally, the export fluxes could not be quantified. The two sole sampling campaign of the river (two periods) are clearly insufficient to measure the export fluxes. The temporal frequency should be at least ten times higher

Experimental design

Insufficient.

Validity of the findings

Unlike what is claimed by authors, the export fluxes could not be quantified. The two sole sampling campaign of the river (two periods) are clearly insufficient to measure the export fluxes. The temporal frequency should be at least ten times higher

Additional comments

The manuscript by Jie Zeng et al. addresses environmentally important question and the general topics is pertinent to the profile of the journal. However, the study has little added value as neither the controlling factors nor the fluxes of metals could not be determined. The title therefore does not reflect the content of the paper

First, the choice of metals is arbitrary and incomplete, despite the fact that ICP MS was used. The authors did not measure Fe, Al, Zn, Sr.
The DOC and Fe are widely recognized as main carriers of trace metals in river waters. Yet, neither Fe nor DOC was analyzed in this study.
Finally, the export fluxes could not be quantified. The two sole sampling campaign of the river (two periods) are clearly insufficient to measure the export fluxes. The temporal frequency should be at least ten times higher

---

## Round 0.2 · accepted · Accept

Thank you for your efforts in revising your manuscript and responding to the reviewer comments in an appropriate way.